# Spatial Design of Childcare Facilities Based on Biophilic Design Patterns

**Sung Jun Park [1],\* and Hyo Chang Lee [2],\***

[1]   Department of Architectural Engineering, Keimyung University, Daegu 42601, Korea
[2]   Department of Interior Architecture & Built Environment, Yonsei University, Seoul 03722, Korea
\*   Correspondence: sjpark@kmu.ac.kr (S.J.P.); spdlee@yonsei.ac.kr (H.C.L.); Tel.: +82-53-580-5765 (S.J.P.); +82-2-2123-4665 (H.C.L.)

**Abstract:** This study covers the issue of insufficient childcare support, which is part of the combined social problem of low birth rates and aging, and sets the direction for the environmental improvement of childcare facilities. This study aims to find the clues to creating an optimized environment for children in nature, which is a key factor that generally promotes children's physical, cognitive, and social development. In this paper, we conducted a literature review and case study to determine the spatial design characteristics of childcare facilities based on a biophilic design and survey. This study reached five conclusions. Firstly, childcare facilities need a spatial design to have a view of the natural ecosystem outside to increase children's concentration and provide a pleasant environment. Secondly, there is a need for open space that makes observation and monitoring more convenient in the different spaces of childcare facilities. Thirdly, childcare facilities need a spatial design where children can enjoy various sensory experiences related to nature. Fourthly, childcare facilities must have an interesting and familiar spatial design using natural elements. Lastly, there must be hiding places considering the children's stage of development and learning ability. The results will be used as the baseline data for the spatial design and planning of childcare facilities based on biophilic design.

**Keywords:** spatial design; childcare facility; biophilic design patterns; childcare; facility planning; biophilia

## 1. Introduction

Social problems caused by low birth rates and an aging population are issues that must be resolved, and it is important to discuss how to solve these problems in a sustainable and human-friendly perspective. In particular, countless studies were conducted on aging to solve socioeconomic problems caused by it throughout the world [1–6]. Aging is actively discussed because the elderly population is now the largest portion of the entire population, which shows promising prospects for related industries and markets that are becoming more subdivided. The increase in the elderly population is caused by the development of science and medical technology, and the extension of the human lifespan led to the realization by many elders that healthy and active aging are the keys to a successful elderly life. Therefore, the government, research institutes, and businesses are implementing various policies, research, and projects to prepare for the aging society. On the other hand, low birth rates are not discussed or studied as much as aging. Moreover, the increase of single-person households or unmarried people led to active discussions about related industries and studies [7,8]. It is only natural to develop policies and research considering single-person households in light of this social phenomenon and trend, and this must be discussed and carried out more actively.

According to the Korea Institute for Health and Social Affairs (2011), low birth rates are caused by job insecurity, increased burden of childcare, and insufficient childcare support. This study focuses

on insufficient childcare support and seeks desirable alternatives to create an environment for the growth and development of children. Furthermore, because human development is a product of dynamic interaction between inborn characteristics and the environment, it is important to provide an environment in which the inborn genetic factors can be revealed as much as possible, in order to optimize children's development [9].

This study aims to find the clues to creating an optimized environment for children in nature, which is a key factor that generally promotes children's physical, cognitive, and social development. Previous studies on children's growth and development showed that natural elements have positive effects on children by inducing physical and mental changes in various aspects, thereby helping them become happier and healthier [10–14]. In other words, an environment rich in natural elements has a positive influence on activities such as children's display of physical abilities, expression of multiple abilities, division of ego-centered territory, and pursuit of social contact.

It is important to integrate and connect people and nature in the building and architectural environment where we live, and the sustainable design strategy to induce positive changes in people is called "biophilic design". This means the systematic integration and connection of people and nature with the space as the medium, instead of an uncritical connection with nature [15–18]. To seek the direction for spatial design of childcare facilities as the starting point of creating the optimized environment for children, this study analyzed 20 childcare facilities from the perspective of biophilic design and presents a spatial design plan for childcare facilities using the characteristics of biophilic design patterns in childcare facilities.

## 2. Research Method and Scope

This study conducted a literature review and case study to determine the spatial design characteristics of childcare facilities based on biophilic design. The detailed research method and scope are described below.

Firstly, we summarized the positive effects of nature on children, as well as the concept, elements, and patterns of biophilic design, based on the literature review. Secondly, we determined the biophilic design patterns of childcare facilities in Japan based on the case study. Thirdly, we integrated and classified similar or redundant items among biophilic design patterns of childcare facilities in Japan from the case study results to identify the characteristics of biophilic design patterns in childcare facilities. Fourthly, a survey was conducted from 6–29 January 2018 using a questionnaire, by visiting childcare facilities and conducting an importance evaluation of 214 guardians of children. Frequency analysis, mean comparison, and factor analysis were conducted on the survey results and collected data using SPSS.

## 3. Literature Review

### 3.1. Childcare Facilities and Natural Elements

Children are physically and mentally immature and grow into human beings with a personality, as they are constantly influenced by surrounding stimulations [19]. In particular, early childhood is the time when basic concepts and attitudes toward the world are formed. According to developmental psychologists, positive attitudes toward certain objects are formed around the age of four. Thus, it is important to educate children from an early age that nature is something that coexists with humans and is highly valued [20,21].

Children show respect and awe for all living things by interacting with those things as they play with nature, which stimulates their infinite imagination and creative associations. They form a comprehensive relationship with nature through this direct encounter and can enjoy an abundant life in early childhood while also developing the refinement necessary for coexisting with nature [22,23].

However, currently, the opportunities for many children to experience nature sharply decreased due to the reduction of outdoor activities and exercises, the increase of using computers and media

in indoor space, and the commuting to school by car [24]. These natural deficits in children also affect in the rapid increase of attention deficit hyperactivity disorder (ADHD) children. In order to alleviate ADHD symptoms, it is necessary to create opportunities to interact with nature in the external space [25].

　　Therefore, a nature-friendly environment that implements various natural elements such as lights, plants, water, soil, and rocks at childcare facilities, which are the main education and play areas for children, promotes growth of peaceful, cooperative, and well-rounded people and has a positive effect on creative and active growth [26]. In particular, it provides a pleasant and comfortable living and educational space that supports children's physical, mental, social, and cognitive development through the mutual formation of community [27].

　　This study conducted a literature review on the positive effects of nature on children, a summary of which is shown in Table 1. Considering the positive effects of nature on children, childcare facilities must have an environment that actively implements natural elements embracing both the body and soul of children.

**Table 1.** Positive effects of nature on children.

| Literature | Position |
|---|---|
| P1, P2 [28,29] | Children's indifference and fear disappear as they spend more time with nature, and they come to enjoy encountering and loving nature. |
| P3 [29,30] | The open space of nature makes children feel free and relaxed, and the encounter with living things in nature provides a sense of mystery and amazement. |
| P4 [14] | Children start to have in-depth questions and curiosity by observing nature, and they show a more enthusiastic attitude toward nature to satisfy their curiosity. |
| P5 [31], P6 [32] | When provided with various experiences of exploration and inspective teaching–learning, children develop scientific thinking and conceptions about nature. |
| P7 [17] | Children perceive countless concepts about the world of nature and promote their understanding as they experience nature. |
| P8 [29] | Children understand the mutual effects of nature and humans by perceiving the interdependence between both, and they understand the human behaviors that damage nature and find solutions. |
| P9 [33] | Through intimacy with nature, children show emotional responses such as love for, a sense of freedom and safety in, and unity with nature. |
| P10 [34] | The intimacy and intellectual curiosity about nature increases the level of interest in nature, thereby inducing environmentally friendly behaviors. |

### 3.2. Concept and Patterns of Biophilic Design

　　Biophilic design is a design concept and standard that aims to actively use various elements and changes of nature in architectural and environmental design. Moreover, the purpose of biophilic design is to integrate and use the characteristics of various natural elements in the architectural environment to expose people to nature and inspire them to coexist with it [17]. Therefore, biophilic design is a sustainable design strategy that integrates and links people and nature. This reduces the emotional shock caused by artificial architectural environments, which complements the sustainable architecture of the past, but does not mean a blind connection with nature [35]. Yet, with a focus on nature's many positive effects on people, biophilic design is based on the hypothesis of "biophilia" that natural elements may have more positive effects on people as they are integrated with the urban environment [36]. These elements include green walls, green roofs, indoor gardens, nature-friendly pedestrian environments, water bodies, etc.

　　Table 2 shows the biophilic design patterns shown in previous studies [37]. Biophilic design elements can be categorized into direct experience of nature, indirect experience of nature, and experience of space and place, which are represented by 11 detailed elements. Moreover, biophilic design patterns can be classified into nature inside space, presenting nature, and spatial characteristics, totaling nine detailed patterns.

**Table 2.** Patterns and definitions of biophilic design.

| Biophilic Design Pattern | | Definition |
|---|---|---|
| Nature inside space | Visual Connection with Nature | Providing views of indoor and outdoor nature, ecosystem, and changes. |
| | Non-Visual Connection with Nature | Providing auditory, olfactory, tactile, and gustatory stimulations from nature, ecosystem, and their changes. |
| | Dynamic and Diffuse Light | Providing a vibrant nature-like environment through various lights and shadows. |
| | Connection with Natural Systems | Providing an environment to feel the healthy changes of nature such as seasonal variations. |
| Presenting nature | Biomorphic Forms and Patterns | Providing symbolism of forms, patterns, materials, and ratios observed in nature. |
| | Material Connection with Nature | Material connection with nature, by minimizing processing and presenting ecological characteristics of the region. |
| | Complexity and Order | Providing an environment to receive various kinds of sensory information of nature focusing on the hierarchy of natural elements. |
| Spatial characteristics | Prospect | Providing an open environment to observe, view, and monitor the surrounding environment. |
| | Refuge | Providing a space that provides a sense of security of being protected from environmental changes. |

In addition, Table 3 shows a summary of research findings about biophilic design patterns and biological responses [37]. Research findings about the biological responses according to biophilic design patterns generally show that there are positive effects on stress relief, cognitive skills, and sensitivity, atmosphere, and preference. In particular, the findings proved that there are effects on increasing preference for space, relieving stress, increasing happiness, and promoting concentration.

**Table 3.** Research findings about biophilic design patterns and biological responses.

| Biophilic Design Pattern | Biological Response | | |
|---|---|---|---|
| | Stress Relief | Cognitive Skills | Sensitivity, Atmosphere, and Preference |
| Visual Connection with Nature | Lowering blood pressure and decreasing heart rate | Increasing work engagement and concentration | Positive effect on spatial preference |
| Non-Visual Connection with Nature | Lowering systolic blood pressure and relieving stress | Positive effect on cognitive skills | Promoting mental health and psychological composure |
| Dynamic and Diffuse Light | Increasing comfort, happiness, and productivity | Positive effect on increased concentration | Improving spatial preference and perception |
| Connection with Natural Systems | Relieving stress | - | Changing the perception about environment and promoting health |
| Biomorphic Forms and Patterns | - | - | Increasing preference for view |
| Material Connection with Nature | - | Lowering diastolic pressure | Promoting relaxed feeling |
| Complexity and Order | Relieving psychological stress | - | Increasing preference for view |
| Prospect | Relieving stress | Reducing boredom and fatigue | Improving ability to perceive comfort and safety |
| Refuge | - | Improving concentration and ability to perceive safety | Improving ability to perceive stability and safety |

## 4. Case Study on Biophilic Design of Childcare Facilities in Japan

This study selected 20 childcare facilities in Japan for a case study to determine the characteristics of biophilic design patterns in childcare facilities. Table 4 shows information about 15 public and five private childcare facilities with a building area of 500 m$^2$ or above.

**Table 4.** General information of childcare facilities in Japan studied in the case.

| Name of Childcare Facility | Location | Building Area (m$^2$) | Feature |
|---|---|---|---|
| NFB Nursery | Nara, Japan | 871.0 | Public childcare facility |
| KM Kindergarten and Nursery | Osaka, Japan | 799.2 | Public childcare facility |
| ST Nursery | Saitama, Japan | 768.7 | Public childcare facility |
| Fukumasu Kindergarten | Chiba, Japan | 512.4 | Public childcare facility |
| TN Nursery | Mie, Japan | 1247.4 | Public childcare facility |
| OA Kindergarten | Saitama, Japan | 647.0 | Public childcare facility |
| AM Kindergarten and Nursery | Kagoshima, Japan | 941.8 | Public childcare facility |
| AN Kindergarten | Kanagawa, Japan | 1386.0 | Public childcare facility |
| SM Nursery | Tokyo, Japan | 622.3 | Private childcare facility |
| Yutaka Kindergarten | Saitama, Japan | 812.0 | Public childcare facility |
| Takeno Nursery | Hyogo, Japan | 512.0 | Public childcare facility |
| D1 Kindergarten and Nursery | Kumamoto, Japan | 1161.0 | Private childcare facility |
| SP Nursery | Fukushima, Japan | 1161.6 | Public childcare facility |
| CO Kindergarten and Nursery | Hiroshima, Japan | 940.6 | Public childcare facility |
| OB Kindergarten and Nursery | Nagasaki, Japan | 864.4 | Public childcare facility |
| Hanazono Kindergarten | Okinawa, Japan | 1107.0 | Public childcare facility |
| Hakusui Nursery School | Chiba, Japan | 530.0 | Public childcare facility |
| Coby Kindergarten | Tokyo, Japan | 1542.0 | Private childcare facility |
| Mokumoku Kindergarten | Tokyo, Japan | 1665.2 | Private childcare facility |
| Leimond-Shonaka Nursery | Aichi, Japan | 1018.0 | Private childcare facility |

This study determined biophilic design elements and patterns shown in various spaces of these facilities, such as the entrance and pedestrian area, playroom, education room, nursing room, restroom, office, outdoor play area, and landscape area. The results are shown in Table 5.

**Table 5.** Characteristics of biophilic design elements and patterns in childcare facilities of Japan.

| Biophilic Design Pattern | | Characteristics of Biophilic Design Elements and Patterns | | | |
|---|---|---|---|---|---|
| Nature inside space | Visual Connection with Nature | <br>NFB Nursery<br><br>Providing an environment to see the changes of the natural ecosystem inside and outside the space in the cafeteria and study room where they spend a lot of time. | <br>KM Kindergarten and Nursery<br><br>Providing an environment to see the changes of the natural ecosystem inside and outside the space in the cafeteria and study room where they spend a lot of time. | <br>OA Kindergarten<br><br>Providing an environment to see the natural ecosystem through the superposition of open space even inside, such as in the playroom and nursing room. | <br>TAKENO Nursery<br><br>Providing an environment to see the external natural ecosystem from anywhere in the childcare facility that is centered on the courtyard. |
| | | <br>D1 Kindergarten and Nursery<br><br>Providing space that offers both the view and experience of the external ecosystem by combining the lower space with the outside. | <br>Hanazono Kindergarten<br><br>Providing space to see the natural ecosystem as much as possible by opening the exterior space of the playroom extensively. | <br>KM Kindergarten and Nursery<br><br>Forming space to see the external natural ecosystem from the study room to increase concentration and secure comfort. | <br>OB Kindergarten and Nursery<br><br>Forming space to see the external natural ecosystem by organizing a large open space integrating the playroom/cafeteria. |

**Table 5.** *Cont.*

| Biophilic Design Pattern | | Characteristics of Biophilic Design Elements and Patterns | | | |
|---|---|---|---|---|---|
| | Non-Visual Connection with Nature |  TN Nursery |  AM Kindergarten and Nursery |  SM Nursery |  D1 Kindergarten and Nursery |
| | | Providing tactile and auditory stimulation from the natural ecosystem by installing water bodies, such as a waterway around the playroom. | Providing tactile and auditory stimulation from the natural ecosystem by installing water bodies, such as a waterway around the playroom. | Providing tactile and auditory stimulation from the complex natural environment such as the natural ecosystem using water and water bodies of gravel. | Forming space that provides tactile and auditory stimulation of play and the natural ecosystem by installing the playroom and water bodies together. |
| | Non-Visual Connection with Nature |  Yutaka Kindergarten |  SP Nursery |  Mokumoku Kindergarten |  AN Kindergarten |
| | | Providing observation of the natural ecosystem and tactile stimulation from natural elements by installing a sand playground in exterior space. | Providing an environment for tactile stimulation from natural elements by installing a sand playground in interior space. | Providing an environment for tactile stimulation from natural elements by installing a sand playground in interior space. | Providing tactile stimulation from nature by finishing and installing play facilities using natural materials. |
| Nature inside space | Dynamic and Diffuse Light |  Fukumasu Kindergarten |  CO Kindergarten and Nursery |  OB Kindergarten and Nursery |  AN Kindergarten |
| | | Building a vibrant and warm environment by using natural light in the large open space. | Giving vitality to the natural light flowing inside through various colors of lightwell and forming an interesting space. | Showing visual differentiation and arousing interest by forming warm and intense indirect lighting in the intensive playroom/study room. | Providing a dynamic indoor environment by creating various shadows with architectural elements of interior space and natural light that flows in. |
| | Connection with Natural Systems |  OA Kindergarten |  KM Kindergarten and Nursery |  Hakusui Nursery School |  SM Nursery |
| | | Forming space to feel the changes of nature such as seasons and weather in the hallway, playroom, and various other spaces through the courtyard. | Forming space to feel the changes of nature outside in spaces with strong personal characteristics such as the restroom and intensive playroom/study room. | Forming space to closely feel the changes of nature outside in all spaces of the higher and lower floors. | Building an environment to feel the diverse and healthy changes of nature by differentiating the landscape characteristics of each space. |
| Presenting of nature | Biomorphic Forms and Patterns |  AM Kindergarten and Nursery |  Hakusui Nursery School |  Yutaka Kindergarten |  Leimond-Shonaka Nursery |
| | | Forming play facilities and areas with sculptures symbolizing natural materials and forms such as trees and forests. | Forming play facilities and areas with sculptures symbolizing natural materials and forms such as trees and forests. | Dividing spaces such as the playroom, study room, and nursing room with sculptures symbolizing natural patterns such as mountains and waves. | Forming entrances and exits and dividing spaces such as the playroom, study room, and nursing room with sculptures symbolizing caves. |

**Table 5.** *Cont.*

| Biophilic Design Pattern | | Characteristics of Biophilic Design Elements and Patterns | | | |
|---|---|---|---|---|---|
| | Material Connection with Nature |  |  |  |  |
| | | Hakusui Nursery School | Takeno Nursery | Takeno Nursery | TN Nursery |
| | | Forming the playroom, study room, and nursing room using natural materials and colors. | Forming architectural elements such as pillars and beams using natural materials representing natural characteristics such as materials and colors. | Forming building facade of the childcare facility using natural materials representing natural characteristics such as materials and colors. | Forming friendly and interesting space by organizing childcare support space for office work and affairs with natural materials. |
| Presenting of nature | Material Connection with Nature |  |  |  |  |
| | | Hanazono Kindergarten | Mokumoku Kindergarten | Mokumoku Kindergarten | Leimond-Shonaka Nursery |
| | | Forming friendly and interesting space by organizing childcare support space for office work and affairs with natural materials. | Using finishing materials expressing natural forms, materials, and patterns, such as annual tree rings, by minimizing processing of natural materials. | Creating an interesting spatial atmosphere by installing sculptures symbolizing shapes and materials of nature such as trees. | Creating a warm and friendly spatial atmosphere by finishing the floors, walls, and ceilings with natural materials. |
| | Complexity and Order |  |  |  |  |
| | | KM Kindergarten and Nursery | Hanazono Kindergarten | SM Nursery | Takeno Nursery |
| | | Providing an environment to receive all kinds of sensory information of nature in nature-oriented play area such as landscape, green footpath, and play facilities. | Providing an environment to receive all kinds of sensory information of nature in nature-oriented play area such as landscape, green footpath, and play facilities. | Forming space to receive all kinds of sensory information of nature in both lower and higher floors through nature-oriented play area and green footpath. | Forming space to receive all kinds of sensory information of nature in both lower and higher floors through nature-oriented play area and green footpath. |
| Spatial characteristics | Prospect |  |  |  |  |
| | | ST Nursery | Hanazono Kindergarten | NFB Nursery | OB Kindergarten and Nursery |
| | | Forming open space for mutual observation and view of each space such as exterior play area, playroom, nursing room, study room, and hallway. | Forming open space for mutual observation and view of each space such as exterior play area, playroom, nursing room, study room, and hallway. | Forming open space for mutual observation and view of each space such as exterior play area, playroom, nursing room, study room, and hallway. | Forming space for observation and view of each room of higher and lower floors and exterior space through large open space. |
| | |  |  |  |  |
| | | OA Kindergarten | ST Nursery | OB Kindergarten and Nursery | Hanazono Kindergarten |
| | | Forming space to observe, view, and monitor main entrance and surrounding exterior space from the interior space. | Forming a waiting room for guardians where they can observe and view the playroom, education room, study room, nursing room, and exterior natural space. | Forming windows for mutual vertical and horizontal observation and view of play area to stimulate curiosity. | Forming windows for mutual vertical and horizontal observation and view of play area to stimulate curiosity. |

| Biophilic Design Pattern | Characteristics of Biophilic Design Elements and Patterns | | | |
|---|---|---|---|---|
| Refuge |  |  |  |  |
| | NFB Nursery | AM Kindergarten and Nursery | AN Kindergarten | Mokumoku Kindergarten |
| | Forming hiding places that are connected to move around and stay to enable physical activities and massed learning. | Forming hiding places that are connected to move around and stay to enable physical activities and massed learning. | Forming hiding places in well-traveled spaces for easy access and view of open interior and exterior space | Forming hiding places using natural forms of sculptures to see the interior and exterior space and arouse interest. |

## 5. Survey Results

### 5.1. General Characteristics of Survey Respondents

This study set up 40 survey items in nine patterns by organizing the characteristics of biophilic design patterns of the childcare facilities shown in the case study. In particular, the questionnaire included the photographs of each case along with the survey items to promote the respondents' understanding.

The importance evaluation was performed by the guardians of the children in childcare facilities. There were 214 respondents with a high ratio of women (89.3%), aged between 30–39 (65.9%), with one child (53.3%) who was between five and six years old (36.9%). Table 6 shows the general characteristics of survey respondents.

**Table 6.** General characteristics of survey respondents.

| Category | Item | N | % |
|---|---|---|---|
| Gender | Male | 23 | 10.7 |
| | Female | 191 | 89.3 |
| | Total | 214 | 100 |
| Age | Age 20–29 | 68 | 31.8 |
| | Age 30–39 | 141 | 65.9 |
| | Age 40 or older | 5 | 2.3 |
| | Total | 214 | 100 |
| No. of children | 1 | 114 | 53.3 |
| | 2 | 94 | 43.9 |
| | 3 or more | 6 | 2.8 |
| | Total | 214 | 100 |
| Child's age * | Age 3–4 | 118 | 36.9 |
| | Age 5–6 | 114 | 35.6 |
| | Age 7 | 88 | 27.5 |
| | Total | 320 | 100 |

* Multiple response.

### 5.2. Importance of Biophilic Design Patterns in Childcare Facilities

Table 7 shows the importance evaluation of guardians about biophilic design in childcare facilities. The total mean of the importance evaluation was 3.84. The importance of biophilic design patterns

above the total mean was highest for "prospect" (4.22), followed by "visual connection with nature" (4.18), "complexity and order" (4.12), and "dynamic and diffuse light" (3.94).

**Table 7.** Importance of biophilic design patterns in childcare facilities *.

| Biophilic Design Pattern | Biophilic Design Pattern Characteristics in Childcare Facilities | Mean of Importance | |
|---|---|---|---|
| Visual Connection with Nature | Providing an environment to see the changes of the natural ecosystem inside and outside the space in the cafeteria and study room where they spend a lot of time. | 4.21 | |
| | Providing an environment to see the natural ecosystem through the superposition of open space even inside such as in the playroom and nursing room. | 4.23 | |
| | Providing an environment to see the external natural ecosystem from anywhere in the childcare facility that is centered on the courtyard. | 4.13 | |
| | Providing space that offers both the view and experience of the external ecosystem by combining the lower space with the outside. | 4.05 | 4.18 |
| | Providing space to see the natural ecosystem as much as possible by opening the exterior space of the playroom extensively. | 4.23 | |
| | Forming space to see the external natural ecosystem from the study room to increase concentration and secure comfort. | 4.34 | |
| | Forming space to see the external natural ecosystem by organizing a large open space integrating the playroom/cafeteria. | 4.06 | |
| Non-Visual Connection with Nature | Providing tactile and auditory stimulation from the natural ecosystem by installing water bodies, such as a waterway around the playroom | 3.85 | |
| | Providing tactile and auditory stimulation from the complex natural environment such as the natural ecosystem using water and water bodies of gravel. | 3.92 | |
| | Forming space that provides tactile and auditory stimulation of play and the natural ecosystem by installing the playroom and water bodies together. | 3.85 | 3.80 |
| | Providing observation of the natural ecosystem and tactile stimulation from natural elements by installing a sand playground in exterior or interior space. | 3.72 | |
| | Providing tactile stimulation from nature by finishing and installing play facilities using natural materials. | 3.64 | |
| Dynamic and Diffuse Light | Building a vibrant and warm environment by expanding the changing natural light to large open space. | 4.16 | |
| | Giving vitality to the natural light flowing inside through various colors of lightwell and forming an interesting space. | 3.73 | 3.89 |
| | Showing visual differentiation and arousing interest by forming warm and intense indirect lighting in the intensive playroom/study room. | 4.17 | |
| | Providing a dynamic indoor environment by creating various shadows with architectural elements of interior space and natural light that flows in. | 3.48 | |
| Connection with Natural Systems | Forming space to feel the changes of nature such as seasons and weather in the hallway, playroom, and various other spaces through the courtyard. | 3.89 | |
| | Forming space to feel the changes of nature outside in spaces with strong personal characteristics such as the restroom and intensive playroom/study room. | 4.16 | 3.77 |
| | Forming space to closely feel the changes of nature outside in all spaces of the higher and lower floors. | 3.72 | |
| | Building an environment to feel the diverse and healthy changes of nature by differentiating the landscape characteristics of each space. | 3.31 | |
| Biomorphic Forms and Patterns | Forming play facilities and areas with sculptures symbolizing natural materials and forms such as trees and forests. | 3.60 | |
| | Dividing spaces such as the playroom, study room, and nursing room with sculptures symbolizing natural patterns such as mountains and waves. | 3.42 | 3.47 |
| | Forming entrances and exits and dividing spaces such as the playroom, study room, and nursing room with sculptures symbolizing caves. | 3.40 | |

**Table 7.** *Cont.*

| Biophilic Design Pattern | Biophilic Design Pattern Characteristics in Childcare Facilities | Mean of Importance | |
|---|---|---|---|
| Material Connection with Nature | Forming the playroom, study room, and nursing room using natural materials representing natural characteristics such as materials and colors. | 3.28 | 3.42 |
| | Forming architectural elements such as pillars and beams using natural materials representing natural characteristics such as materials and colors. | 3.45 | |
| | Forming building facade of the childcare facility using natural materials representing natural characteristics such as materials and colors. | 3.34 | |
| | Forming friendly and interesting space by organizing childcare support space for office work and affairs with natural materials. | 3.79 | |
| | Using finishing materials expressing natural forms, materials, and patterns, such as annual tree rings, by minimizing processing of natural materials. | 3.26 | |
| | Creating an interesting spatial atmosphere by installing sculptures symbolizing shapes and materials of nature such as trees. | 3.43 | |
| | Creating a warm and friendly spatial atmosphere by finishing the floors, walls, and ceilings of space with natural materials. | 3.41 | |
| Complexity and Order | Providing an environment to receive all kinds of sensory information of nature in nature-oriented play area such as landscape, green footpath, and play facilities. | 4.18 | 4.12 |
| | Forming space to receive all kinds of sensory information of nature in both lower and higher floors through nature-oriented play area and green footpath. | 4.05 | |
| Prospect | Forming open space for mutual observation and view of each space such as exterior play area, playroom, nursing room, study room, and hallway. | 4.47 | 4.22 |
| | Forming space for observation and view of each room of higher and lower floors and exterior space through large open space. | 4.22 | |
| | Forming space to observe, view, and monitor main entrance and surrounding exterior space from the interior space. | 4.30 | |
| | Forming a waiting room for guardians where they can observe and view the playroom, education room, study room, nursing room, and exterior natural space. | 4.13 | |
| | Forming windows for mutual vertical and horizontal observation and view of play area to stimulate curiosity. | 3.96 | |
| Refuge | Forming hiding places that are connected to move around and stay to enable physical activities and massed learning. | 3.85 | 3.76 |
| | Forming hiding places in moving spaces for easy access and view of open interior and exterior space. | 3.60 | |
| | Forming hiding places using natural forms of sculptures to see the interior and exterior space and arouse interest. | 3.84 | |
| Total mean | | 3.84 | |

* Five-point scale; the shaded parts are items with 4.0 or higher importance.

As for the mean of importance by item, the highest was "forming open space for mutual observation and view of each space such as exterior play area, playroom, nursing room, study room, and hallway" (4.47), followed by "forming space to see the external natural ecosystem from the study room to increase concentration and secure comfort" (4.34), "forming space to observe, view, and monitor main entrance and surrounding exterior space from the interior space" (4.30), and "providing an environment to see the natural ecosystem through the superposition of open space even inside such as in the playroom and nursing room".

The overall biophilic design pattern characteristics of childcare facilities with high importance were "prospect" and "visual connection with nature", by facilitating observation and view of the natural ecosystem inside and outside by opening each space and by creating a monitoring environment of the entrance. Furthermore, it was important to create a friendly and interesting environment through nature-oriented play areas and dynamic light environments for children to receive all kinds of sensory information. However, biophilic design pattern characteristics of childcare facilities which showed relatively low importance in the analysis result, such as the application of materials expressing nature in indoor space and the shelter of natural form, should be considered in planning childcare facilities. This is because, in the case of childcare facilities in urban areas, it is difficult to create an environment that satisfies the detailed characteristics of the biophilic design patterns that are highly important due to the urban context.

### *5.3. Factors and Characteristics of Childcare Facilities Based on Biophilic Design*

Table 8 shows the factor analysis based on the results of the guardians' importance evaluation to determine the characteristics of childcare facilities based on biophilic design.

The factor analysis was conducted to provide a spatial design plan based on biophilic design in childcare facilities that guardians consider important. The results showed that all elements are categorized into four factors, and the explanatory power of all factors (rotation sums of squared loading: cumulative percentage) was 72.846%. Factors classified by the factor analysis were Factor 1 ("environment to easily observe, view, and monitor"), Factor 2 ("interesting and familiar environment"), Factor 3 ("environment to experience various kinds of sensory information"), and Factor 4 ("environment to support intensive play and learning"). The mean of importance was highest for Factor 4 ("environment to support intensive play and learning") (4.16), followed by Factor 1 ("environment to easily observe, view, and monitor") (4.12), Factor 3 ("environment to experience various kinds of sensory information") (3.94), and Factor 2 ("interesting and familiar environment") (3.64).

**Table 8.** Factor analysis results of biophilic design pattern characteristics in childcare facilities *.

| Factor | Biophilic Design Pattern Characteristics in Childcare Facilities | Component | Mean of Importance | |
|---|---|---|---|---|
| Environment to easily observe, view, and monitor | Forming open space for mutual observation and view of each space such as exterior play area, playroom, nursing room, study room, and hallway. | 0.875 | 4.47 | |
| | Forming space to observe, view, and monitor main entrance and surrounding exterior space from the interior space. | 0.854 | 4.30 | |
| | Providing an environment to see the external natural ecosystem from anywhere in the childcare facility that is centered on the courtyard. | 0.802 | 4.13 | |
| | Providing space to see the natural ecosystem as much as possible by opening the exterior space of the playroom extensively. | 0.756 | 4.23 | |
| | Forming space for observation and view of each room of higher and lower floors and exterior space through large open space. | 0.724 | 4.22 | 4.12 |
| | Forming a waiting room for guardians where they can observe and view the playroom, education room, study room, nursing room, and exterior natural space. | 0.688 | 4.13 | |
| | Forming space to see the external natural ecosystem by organizing a large open space integrating the playroom/cafeteria. | 0.612 | 4.06 | |
| | Providing space that offers both the view and experience of the external ecosystem by combining the lower space with the outside. | 0.541 | 4.05 | |
| | Forming space to feel the changes of nature such as seasons and weather in the hallway, playroom, and various other spaces through the courtyard. | 0.501 | 3.89 | |
| | Forming space to closely feel the changes of nature outside in all spaces of the higher and lower floors. | 0.487 | 3.72 | |
| Interesting and familiar environment | Building a vibrant and warm environment by expanding the changing natural light to large open space. | 0.867 | 4.16 | |
| | Forming windows for mutual vertical and horizontal observation and view of play area to stimulate curiosity. | 0.815 | 3.96 | |
| | Forming hiding places using natural forms of sculptures to see the interior and exterior space and arouse interest. | 0.798 | 3.84 | |
| | Giving vitality to the natural light flowing inside through various colors of lightwell and forming an interesting space. | 0.716 | 3.73 | |
| | Forming play facilities and areas with sculptures symbolizing natural materials and forms such as trees and forests. | 0.687 | 3.60 | 3.61 |
| | Forming hiding places in moving spaces for easy access and view of open interior and exterior space. | 0.621 | 3.60 | |
| | Dividing spaces such as the playroom, study room, and nursing room with sculptures symbolizing natural patterns such as mountains and waves. | 0.597 | 3.42 | |
| | Forming entrances and exits and dividing spaces such as the playroom, study room, and nursing room with sculptures symbolizing caves. | 0.496 | 3.40 | |
| | Creating an interesting spatial atmosphere by installing sculptures symbolizing shapes and materials of nature such as trees. | 0.412 | 3.43 | |
| | Creating a warm and friendly spatial atmosphere by finishing the floors, walls, and ceilings of space with natural materials. | 0.369 | 3.41 | |
| | Using finishing materials expressing natural forms, materials, and patterns such as annual tree rings by minimizing processing of natural materials. | 0.301 | 3.26 | |

**Table 8.** *Cont.*

| Factor | Biophilic Design Pattern Characteristics in Childcare Facilities | Component | Mean of Importance |
|---|---|---|---|
| Environment to experience various kinds of sensory information | Providing an environment to receive all kinds of sensory information of nature in nature-oriented play area such as landscape, green footpath, and play facilities. | 0.767 | 4.18 |
| | Forming space to receive all kinds of sensory information of nature in both lower and higher floors through nature-oriented play area and green footpath. | 0.715 | 4.05 |
| | Providing tactile and auditory stimulation from the complex natural environment such as the natural ecosystem using water and water bodies of gravel. | 0.654 | 3.92 |
| | Forming space that provides tactile and auditory stimulation of play and the natural ecosystem by installing the playroom and water bodies together. | 0.541 | 3.85 |
| | Providing tactile and auditory stimulation from the natural ecosystem by installing water bodies such as a waterway around the playroom. | 0.502 | 3.85 |
| | Providing observation of the natural ecosystem and tactile stimulation from natural elements by installing a sand playground in exterior or interior space. | 0.425 | 3.72 |
| | Providing tactile stimulation from nature by finishing and installing play facilities using natural materials. | 0.311 | 3.64 |
| Environment to support intensive play and learning | Forming space to see the external natural ecosystem from the study room to increase concentration and secure comfort. | 0.856 | 4.34 |
| | Providing an environment to see the natural ecosystem through the superposition of open space even inside such as in the playroom and nursing room. | 0.795 | 4.23 |
| | Showing visual differentiation and arousing interest by forming warm and intense indirect lighting in the intensive playroom/study room. | 0.695 | 4.17 |
| | Providing an environment to see the changes of the natural ecosystem inside and outside the space in the cafeteria and study room where they spend a lot of time. | 0.601 | 4.21 |
| | Forming space to feel the changes of nature outside in spaces with strong personal characteristics such as the restroom and intensive playroom/study room. | 0.523 | 4.16 |
| | Forming hiding places that are connected to move around and stay to enable physical activities and massed learning. | 0.412 | 3.85 |

*Values of Mean of Importance for grouped factors: 3.94 (Environment to experience various kinds of sensory information), 4.16 (Environment to support intensive play and learning).*

\* Factor extraction method: principle component analysis; rotation method: varimax with Kaiser; normalization: rotation in 17 iterations; the shaded parts are items with 4.0 or higher importance.

Each factor of the factor analysis is a type of an important spatial design plan for applying biophilic design to childcare facilities, and the characteristics of biophilic design patterns in each type are the detailed items. Firstly, "environment to easily observe, view, and monitor" can be planned as a space that can be viewed and observed through direct or indirect connection with nature in the inside or outside in building. Therefore, it is possible for children to observe the flow of time and season through transparent materials of the opening and the positioning of the openings. Also, it is significantly important that the openness of the indoor space is planned so that children's activities can be monitored. Secondly, "interesting and familiar environment" is to stimulate children's curiosity through the recognizing the change of nature in indoor space and to plan an environment that gives the opportunity to be familiar with nature through elements such as light, water, etc. Thirdly, "environment to experience various kinds of sensory information" needs a plan to stimulate children's five senses through facilities and spaces that provide opportunities to experience natural elements in outer space and inner space. Fourthly, "environment to support intensive play and learning" should be considered an environmental plan that enables offering visual stimulation through detecting changes of nature in children's learning and play space and the visual differentiation through artificial lighting, etc. Therefore, the important features for the spatial design of childcare facilities based on biophilic design that actively use natural elements are as follows: creating an open environment to observe, view, and monitor the surroundings; forming an interesting and familiar space for intensive play and learning by using natural elements and securing the view of the natural ecosystem; and using natural elements for various visual and non-visual experiences.

## 6. Conclusions and Discussion

This study focused on the issue of insufficient childcare support, which is part of the combined social problem of low birth rates and aging, and sets the direction for the environmental improvement

of childcare facilities. Furthermore, the fact that this study pursued improvement from the theory of biophilic design patterns, which prefers nature and life, in setting the direction for childcare facilities differentiates this study from others. The results will be used as the baseline data for the spatial design and planning of childcare facilities based on biophilic design. Based on the results, this study reached the five conclusions outlined below.

Firstly, childcare facilities need a spatial design to have a view of the natural ecosystem outside to increase children's concentration and provide a pleasant environment. The spatial design must allow users to view the changes of the natural ecosystem to reduce stress in spaces such as playrooms, study rooms, and even hiding places and to improve children's concentration while also securing safety and stability. In other words, architects and interior designers need to find a way to maximize the outdoor natural ecosystem into the indoor environment in planning the childcare facility, and to recognize the relationship between the learning environment and the natural ecosystem and plan the space.

Secondly, there is a need for open space that makes observation and monitoring more convenient in the different spaces of childcare facilities. This study presents five ways to secure open space in childcare facilities: visually opening each space of the childcare facility centered on the courtyard, securing vision through opening and overlapingeach space away from the outside, facilitating observation of both higher and lower floors by extensively integrating the playroom and cafeteria, facilitating observation between interior and exterior spaces from the entrance or moving space by including fences, and designing the waiting area for guardians where they can see each space, as well as the natural ecosystem on the outside. Therefore, the planning and space design of the childcare facility should be able to observe and monitor the nature through the openness of the outdoor space and the indoor spatial plan connected with the inside and outside.

Thirdly, childcare facilities need a spatial design where children can enjoy various sensory experiences related to nature. This includes visual, tactile, auditory, and olfactory experiences, for which it is necessary to apply natural light, morphological imitation of natural elements, and texture of natural elements to the spatial design of childcare facilities. In particular, it is necessary to design the outdoor play area for children to have various sensory experiences of the seasonal changes of nature. To this end, there must be a combination of landscape and play equipment, green pedestrian paths, and water bodies made up of water and gravel. It is also necessary to consider building a sand playground inside or outside for play and tactile experiences for children. The element of experience that induces various senses for children can be found in nature, and, although there is a plan to offer a play space outside of the childcare facility, the expression and application of elements that allow experiencing nature with five senses is necessary in indoor space.

Fourthly, childcare facilities must have an interesting and familiar spatial design using natural elements. To this end, there is a need for spatial design that organizes or divides space with sculptures imitating forms of nature such as trees, forests, mountains, waves, and caves, as well as finishing materials that show the textures and patterns of natural elements by minimizing processing. The program of the childcare facility carries out many educational activities in the indoor space rather than the outdoor space. Therefore, it is very important to stimulate children's curiosity by applying shapes, materials, etc. in planning the indoor space.

Lastly, there must be hiding places considering children's stage of development and learning ability. This means a hiding place that can also be observed, that is, not an isolated space but a place made using play equipment or sculptures in the play or recreation area, as well as interior landscape or sculptures in other interior spaces. Spatial design for such a refuge needs to facilitate access and use by locating it in a well-traveled space; it also needs to combine play equipment and refuge so that children can use the areas autonomously for intensive learning and interaction between small groups, and it should arouse interest by imitating forms of nature such as caves and trees.

This study extracted the elements of planning through a case analysis of childcare facilities and dealt with the evaluation of the importance of planning factors for the child caregivers. However, it is a limitation of this study that only 20 cases of childcare facilities in Japan were dealt with in the process

of deriving the planning factors. In further study, detailed research must be conducted on children's satisfaction and importance of the spaces applying biophilic design in childcare facilities considering children's developmental stages.

**Author Contributions:** H.C.L. and S.J.P. conceived and designed the research and wrote the paper; H.C.L. analyzed the data; S.J.P. critically reviewed the article.

**Funding:** This research was funded by National Research Foundation of Korea(NRF) grant funded by the Korea Government (MEST) (NRF-2018R1C1B6008735).

**Conflicts of Interest:** The authors declare no conflicts of interest.

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
