# Peer review of "Spatial Design of Childcare Facilities Based on Biophilic Design Patterns"

_sustainability, doi:10.3390/su11102851_

Reviewer 1 Report

Thank you for the opportunity to review. I believe this paper takes a very interesting and novel perspective in how childcare environments can be designed to promote healthy development among young children. This information is important; however, I have provided some high-level suggestions to help strengthen this submission.

Overall, I think the paper could be better organized. I found it a bit confusing to read.

To me, I think the literature review is a key objective of this study and brings much value (high-level summary of existing literature). However, less attention was placed on this objective in comparison to the singular case study and am curious to why this is the case. Please explain.  And to this end, I think the presented information from the literature review could be further developed – it seems somewhat lacking.

Please include additional details regarding your search strategy as I’m unsure which sources you used to pull the literature for your review. Were grey literature or policy briefs considered? How did you decide to group/synthesize the data you pulled from your lit review (sub-headings, etc.)?

Because the first part of the paper focused on a literature review, I think references need to be included here to direct readers where particular statements or findings were pulled from? Supporting references appear to be missing.

Multiple paragraphs included in the “conclusion” section should eb moved to the “discussion” section.

Please acknowledge the limitations of this work in the “discussion” section.

Respectfully submitted.

Author Response

Dear Reviewer,

We are grateful for the careful review of this paper.
We have revised and supplemented this paper by actively reflecting your opinions.
We have mentioned some reasons why the contents of the revision and supplementation have not been complemented. The modifications and supplements are as the attachment on the next page.
I'd appreciate it if you could need more review this revised manuscript and let us know about additional revision.
Thank you for your review of our manuscript. We look forward to hearing from you.

Sincerely

Reviewer 2 Report

I think this is important research. I have suggestions that can strengthen this paper; 

the opening discussion about aging and low birth rates does not seem relevant to the actual research. Many Western countries turn to immigration to offset low birth rates by the majority population. This is a complex issue and it is not developed in a way that relates well to the environment and space considerations in children's nursery schools. (40,41)

the issues with low birth rates is a national issue - not an issue of the human race - as in #1 above - there are plenty of humans available to make up the shortfall and balance out the demographics in any country - however national policies respond to this issue differently.  (45,46) 

How is it possible, as it is suggested, that the issue of low birth rates can be resolved by providing children's daycare spaces based on connection to nature? (50,51) This doesn't make logical sense - however providing high quality early childhood spaces may encourage people to have more children - but this is a very complex issue.

What does 'blinded" connection to nature mean? perhaps find another way of describing this? (65)

It is difficult to do a study or literature review in this area without citing the work of Richard Louv (nature deficit) and E.O Wilson (biophilia)

I like the charts and the pictures - but what is it about the Japanese approaches to Child care the connects education to nature? Why was Japan chosen and not Korea? 

The number of characteristics and all the values cited (tables 7 & 8) need more discussion other than that provided in the conclusion - What are the implications for designers of child care spaces? Why is this important? There is too much reliance on the statistics and not enough discussion and connecting the research to the implications for educational practice. At the end of the paper  I would suggest a longer section titled "Discussion and Implications" 

Author Response

Dear Reviewer,

We are grateful for the careful review of this paper.
We have revised and supplemented this paper by actively reflecting your opinions.
We have mentioned some reasons why the contents of the revision and supplementation have not been complemented. The modifications and supplements are as the attachment on the next page.
I'd appreciate it if you could need more review this revised manuscript and let us know about additional revision.
Thank you for your review of our manuscript. We look forward to hearing from you.

Sincerely

Round  2

Reviewer 2 Report

The authors have responded adequately to the suggested revisions.